# Botnet Defense System: Observability, Controllability, and Basic Command and Control Strategy

**DOI:** 10.3390/s22239423

**Published:** 2022-12-02

**Authors:** Shingo Yamaguchi

**Affiliations:** Graduate School of Sciences and Technology for Innovation, Yamaguchi University, Ube 755-8611, Japan; shingo@yamaguchi-u.ac.jp

**Keywords:** malicious botnet, white-hat botnet, botnet defense system, observability and controllability, command and control, strategy

## Abstract

This paper deals with the observability, controllability, and command and control strategy in the Botnet Defense System (BDS) that disinfects malicious botnets with white-hat botnets. The BDS defends an IoT system built over the Internet. The Internet is characterized by openness, but not all nodes are observable and controllable. We incorporated the concept of observability and controllability into the BDS design and theoretically clarified that the BDS can enhance its observability and controllability by utilizing its white-hat botnets. In addition, we proposed a Withdrawal strategy as a basic strategy to command and control white-hat botnets. Then, we modeled the BDS, adopted the Withdrawal strategy with agent-oriented Petri net PN^2^ and confirmed the effect through the simulation of the model. The result shows that even if considering observability and controllability, the BDS wiped out the malicious bots and reduced the white-hat bots to less than 1% as long as the white-hat worms were sufficiently infectious.

## 1. Introduction

The Internet of Things (IoT) has made remarkable progress and has brought benefits to our lives. On the other hand, IoT has created new threats to cyber security. In September 2016, the threat became a reality. A lot of IoT devices were misused to cause a massive and destructive Distributed Denial of Service (DDoS) attack. A new kind of malware called Mirai [1] was used to hijack them. Hijacked devices are called bots because they can be controlled remotely. The network of bots is called a botnet and is available to perform various cyber attacks such as DDoS attacks, digital scams and malicious cryptocurrency mining. IoT devices are known to be vulnerable. This is because many of them do not have sufficient resources to perform security functions and vendors may sacrifice security in order to compete on price or rush to market. To make matters worse, Mirai’s authors released the source code [2]. This brought about the birth of many variants such as Okiru and Satori. As cyber space and real space are integrated, IoT botnets have become one of the major threats in today’s society.

To deal with the threat of IoT botnets, there are several techniques such as detection, mitigation and spread prevention. These techniques can help mitigate the threat of botnets, but they cannot disinfect botnets that have spread already. It is known that the Mirai botnet can be removed by rebooting the infected devices because it penetrates only in the dynamic memory of the devices [3]. However, with the explosive growth of IoT devices, it is not practical to go around rebooting the infected devices. With the Botnet Defense System (BDS) [4], it has been proposed to use a botnet instead of humans to disinfect malicious botnets. The BDS builds a botnet itself and uses the botnet to disinfect the malicious botnet.

An IoT system is built on the Internet rather than a closed network. This enables the IoT system to utilize not only physical devices but also various services on the Internet. On the other hand, it is known that the Internet is difficult to monitor and manage due to its heterogeneity, geographic size and distributed management [5]. Therefore, this system configuration may take a risk on cyber security. To resolve this, botnet technology may become a breakthrough. The botnet autonomously spreads and expands even to the nodes unobservable and uncontrollable by the security system. This may enable the security system to use a self-build botnet to indirectly monitor the unobservable and uncontrollable nodes. However, there has been no discussion on the operation of the botnet over the network including unobservable and uncontrollable nodes.

In this paper, we propose a mechanism to command and control a self-build botnet for disinfecting malicious botnets over the network including unobservable and uncontrollable nodes. Our main contributions are as follows.

The concept of observability and controllability was firstly introduced and discussed in the research area of botnet disinfection.The mechanism of command and control was clarified in terms of Petri nets [6] theoretically.A command and control strategy, named Withdrawal strategy, was proposed and the effect was quantitatively confirmed through simulation evaluation.

The rest of this paper is organized as follows: Section 2 introduces related work. Section 3 presents the concept and design of BDS. Section 4 presents the concept of observability and controllability introduced into the BDS and describes the mechanism to command and control the botnet under the observability and controllability. In addition, it provides the strategy of commanding and controling the botnet. Section 5 presents modeling of the proposed mechanism and the strategy with agent-oriented Petri net PN^2^ [7] and shows the effectiveness through the simulation evaluation with the PN^2^ model. Section 6 summarizes our key points and shows open problems to be dealt with in the future.

## 2. Related Work

Several countermeasure techniques have been proposed against botnet threats [8]. These can be broadly classified into three categories: detection, mitigation and spread prevention (see Figure 1 and Table 1).

The detection technique is to decide whether there exists a botnet in the targeted IoT system (see Figure 1i). This is a crucial first step to dealing with this threat. Intrusion Detection System (IDS) is widely used as a detection technique. There are two kinds of approaches to detection: signature-based and anomaly-based. Signature-based detection registers malicious patterns in a database and detects botnets by matching the registered patterns. Said et al. [9] investigated a syntactic method with string-based signatures and a semantic method with system-call-based signatures in the detection of Mirai. The evaluation results show that the semantic method shows better detection rate than the syntactic method and is resistant to simple obfuscation to disguise malicious properties. The signature-based approach has been adopted by open-source IDSs such as Snort [20] and Suricata [21]. On the other hand, anomaly-based detection registers normal patterns in a database and detects botnets by classifying any deviation from the registered patterns. Bezerra et al. [10] proposed a host-based detection method utilizing one-class classifiers. This method focuses on a single device and extracts features from the device’s behavior such as CPU utilization and memory consumption. The evaluation results show that one-class classifiers are enough to detect a different botnet and have a small impact on the device’s utilization and energy consumption. Soe et al. [11] proposed a detection framework with a sequential architecture based on machine learning algorithms. The framework is characterized by adopting a correlated feature selection method to reduce the number of irrelevant features. This enables implementation in resource-constrained IoT devices. Meidan et al. [12] proposed a network-based detection method. This method extracts behavior snapshots of a targeted network and applies deep autoencoders to detect anomalous traffic. The evaluation results show that the proposed method can accurately and instantly detect the attacks launched from a botnet. Hoang et al. [13] proposed a botnet detection model based on machine learning using Domain Name Service query data and confirmed that machine learning algorithms can be used effectively in botnet detection. Mihoub et al. [14] proposed an architecture composed of two components for detecting and mitigating DoS/DDoS attacks. The detection component is a multi-class classifier that adopts the Looking-Back concept and identifies the type of attack and the packet type used in the attack. Based on the result, the mitigation component applies the corresponding countermeasure. Shinan et al. [22] conducted a survey on various techniques for detecting botnets, including botnet attacks, machine learning and Software Defined Networking (SDN). They described graph-based features based on machine learning for bot detection in SDN as the most promising avenue.

The mitigation technique aims to prevent and/or reduce the effect of botnet on the IoT system itself and/or victim systems (see Figure 1ii). It is often used in conjunction with detection techniques to respond to detected botnets. Manso et al. [15] proposed a software-defined IDS. When the system detects an attack, it uses an SDN controller to control the traffic. Ceron et al. [16] proposed an adaptive network layer to block or rewrite the traffic generated by botnets to invalid ones. Specifically, it makes it possible to block or rewrite the instructions sent from the botnet controller to the bots to make them invalid. Khattak et al. [17] proposed an open source tool, BotFlex, with an extensible module system. It aims to continually improve mitigation for botnets across the entire IT community. Researchers and companies can contribute to improving the mitigation by developing BotFlex modules.

The spread prevention technique aims to protect vulnerable devices against botnets (see Figure 1iii). It includes changing default passwords, and the removal of unnecessary software or unnecessary usernames. In Reference [3], the United States Computer Emergency Readiness Team (US-CERT) recommends users and administrators of IoT devices take preventive steps such as changing all default passwords to stronger ones, applying the latest security patches and monitoring the ports that Mirai exploits. Frank et al. [18] developed Python scripts that run on actual devices to prevent IoT devices from becoming Mirai bots. The first hardening script prevents the first infection, while the second script recognizes existing infections and prevents reinfection. Gopal et al. [19] proposed a whitelist-based solution to prevent the spread of Mirai and showed that they successfully blocked the Mirai malware through experiments.

These techniques can help mitigate the threat of botnets but cannot disinfect a botnet which has already spread. The botnet can be removed by rebooting the infected devices. However, as the number of IoT devices increases explosively, it is not practical to deal with them by human tactics. This solution requires an innovative approach.

## 3. Botnet Defense System (BDS)

Botnet Defense System (BDS) [4] is a cyber security system designed to disinfect a botnet that has already spread. The concept of BDS is “Fight Fire with Fire,” i.e., the BDS responds to the malicious botnet with botnet technology. The BDS builds a white-hat botnet itself and uses the botnet to disinfect the malicious botnet. The white-hat botnet can spread autonomously and has the ability to get back devices from the malicious botnet [23]. Therefore, the BDS is expected to drastically increase defense capability against the malicious botnet.

We illustrate the system configuration of the BDS in Figure 2. The left and right rounded rectangles, respectively, represent the BDS and the IoT system defended by the BDS. The BDS repeats the four-step process consisting of monitoring, strategic planning, botnet construction and botnet command and control (C&C) to defend IoT systems from malicious botnets.

Step 1*Monitoring*: Monitor the IoT system and check if the system is infected by any malicious botnet. Once detecting the malicious botnet, the BDS investigates it and collects the information such as its type and spread status.Step 2*Strategy Planning*: Plan strategies for disinfecting the malicious botnet based on the information obtained in Step 1.Step 3*Botnet Building*: Build the white-hat botnet by sending white-hat worms [23] to the IoT systems according to the strategy.Step 4*Botnet C&C*: Command and control the white-hat botnet spreading autonomously for disinfecting the malicious botnet according to the strategy.

The inner graph (N,E) of the right rounded rectangle of Figure 2 represents the network configuration of the IoT system. Each node *n* (∈N), depicted as □, represents a network node, each of which possesses a single device *d*. Each edge *e* (∈E) represents a connection among the network nodes. Different networks have different configurations. In this example, the network consists of 24 nodes $n_{1},n_{2},\cdots ,n_{24}$ and has a square lattice topology. The arc connecting the BDS to a network node represents a connection between the BDS and the node.

The BDS uses the white-hat worm proposed in Reference [23]. The white-hat worm infects and turns a device into a white-hat bot.

Lifespan: The worm is only available for a limited time. It will destroy itself at the end of a given time. That is, the time period means the lifespan of the white-hat worm. Even after the worm self-destructs, the device remains a white-hat bot and is immune until being rebooted.Secondary infectivity: The worm can infect the device that has already been a malicious bot and then clear the malicious bot from the device.

The BDS must build the white-hat botnet so that it has the upper hand over the malicious botnet. However, the white-hat botnet should be built as small as possible because it protects the IoT system but consumes its resources. Therefore, the BDS is required to build the white-hat botnet of an appropriate size depending on the type and spread status of the malicious botnet and the capabilities of available white-hat worms. The BDS can choose a strategy for building the white-hat botnet among the strategies: all-out, few-elite and environment-adaptive proposed in Reference [23].

Let us explain how the BDS works using the state transition shown in Figure 3 as an example. Figure 3a shows the state in which a malicious botnet is invading the IoT system. In Step 1, the BDS detected the malicious botnet and identified ten devices that have already become malicious bots in the system. The malicious bots are depicted as ● in Figure 3a. In Step 2, the BDS decided to adopt the few-elite strategy for disinfecting the malicious botnet because the BDS had a white-hat worm with a strong secondary infectivity. Figure 3b shows the state when the BDS built a white-hat botnet in Step 3. In accordance with the few-elite strategy, the BDS sent the white-hat worm only to two nodes $n_{10},n_{15}$ and built the initial white-hat botnet. The white-hat bots are depicted as ○ in Figure 3b. After that, the malicious botnet and the white-hat botnet autonomously spread over the network. Figure 3c shows the state when the white-hat worm at $n_{15}$ secondarily infected the device $d_{19}$ at node $n_{19}$ that had already been a malicious bot. The malicious bot was cleared from $d_{19}$. However, $d_{19}$ was turned into a white-hat bot instead. The white-hat botnet spread throughout the system while disinfecting the malicious botnet by secondary infection. Figure 3d shows a state in which the white-hat botnet has wiped out the malicious botnet.

## 4. Observability, Controllability, and Command and Control Strategy

The IoT system defended by the BDS is built on the Internet. The Internet is characterized by openness but not all parts of the Internet are communicable. Restricting communication between networks enhances cyber security. However, it may become an obstacle to the operation of cyber security systems including the BDS.

Botnet technology has accelerated its evolution since the release of the Mirai source code and has spawned a variety of variants. The purpose of the variants is no longer limited to DDoS attacks, but also extends to electronic scams, and illicit cryptocurrency mining. Noteworthy is the proxy function provided by a variant called OMG. It enables the infected IoT device to act as a pivot point between networks [24]. Here, it is important to note that the BDS uses botnets. Incorporating such a message transfer function into the white-hat botnet enables the BDS to further enhance the ability of the BDS.

### 4.1. Observability and Controllability by BDS

We introduce the concepts of observability and controllability. In a network defended by the BDS, we define the observability and controllability between the BDS and nodes as follows.

**Definition** **1** (Observability and controllability by BDS).
*In a network defended by the BDS, let n be a node in the network.*

*n is said to be observable by the BDS if the BDS can check whether there is any bot at n. If there is a connection from n to the BDS, the BDS can check whether there is any bot at n.*

*n is said to be controllable by the BDS if the BDS can send a white-hat worm to n. If there is a connection from the BDS to n, the BDS can send a white-hat worm to n.*



Let us illustrate the concept of the observability and controllability by the BDS with an example shown in Figure 4. In this example, the network has the same structure as Figure 3, i.e., it consists of 24 nodes $n_{1},n_{2},\cdots ,n_{24}$ and has a square lattice topology. The difference between them is observability and controllability. In this example, the BDS has connections from and to nodes $n_{1},n_{2},\cdots ,n_{12}$. This means that those nodes are observable and controllable, while the others $n_{13},n_{14},\cdots ,n_{24}$ are unobservable and uncontrollable. We apply a black diagonal lines pattern meshing to the unobservable and uncontrollable nodes in Figure 4. Figure 4a shows the state when the BDS detected the malicious botnet. The BDS can check whether there is a bot only at the observable nodes $n_{1},n_{2},\cdots ,n_{12}$. As a result, the BDS found only 5 of 10 bots at nodes $n_{1},n_{2},n_{4}$, $n_{6}$ and $n_{11}$.

The BDS adopted the few-elite strategy in Step 2 and used two white-hat worms to build the malicious botnet in Step 3. One worm was sent to the same node $n_{10}$ as the former example. The other was sent to another node $n_{7}$ different from the previous example because the BDS can only send it to a controllable node. Figure 4b shows the state when the BDS built a white-hat botnet in Step 3.

### 4.2. Observability and Controllability by White-Hat Bot

The BDS is characterized by using a white-hat botnet to defend the network. Each bot is given the same ability to observe and control network nodes as the BDS. However, the nodes that it can observe and control are limited to its neighbors.

**Definition** **2** (Observability and controllability by white-hat bot).
*In a network (N,E) defended by the BDS, let n and $n^{\prime }$ be nodes in (N,E) where n has a white-hat bot w.*

*$n^{\prime }$ is said to be observable by w if w can check whether there is any bot at $n^{\prime }$. If there is a connection from $n^{\prime }$ to n, i.e., $(n^{\prime },n)\in E$, w can check whether there is any bot at $n^{\prime }$.*

*$n^{\prime }$ is said to be controllable by w if w can send a white-hat worm to $n^{\prime }$. If there is a connection from n to $n^{\prime }$, i.e., $(n,n^{\prime })\in E$, w can send a white-hat worm to $n^{\prime }$.*



Let us consider Figure 4b again. The BDS launched white-hat bots at two controllable nodes: $n_{7}$ and $n_{10}$. For node $n_{7}$, its adjacent nodes $n_{3},n_{6},n_{8},n_{11}$ are observable and controllable nodes by the white-hat bot at node $n_{7}$. On the other hand, for node $n_{10}$, its adjacent nodes $n_{6},n_{9},n_{11},n_{14}$ are observable and controllable nodes by the white-hat bot at node $n_{10}$. Note that $n_{14}$ is unobservable and uncontrollable by the BDS.

### 4.3. Observability and Controllability by BDS via White-Hat Botnet

Some nodes in the network are observable and controllable by the BDS, while others are not. The white-hat botnet can autonomously spread and expand even to nodes unobservable and uncontrollable by the BDS. The BDS can use the spread white-hat botnet to expand the scope that the BDS can observe and control.

The BDS has the ability to exchange messages with a white-hat bot that is at a node observable and controllable by itself. A white-hat bot is also given the same ability as the BDS.

**Definition** **3** (Message exchange ability).
*In a network (N,E) defended by the BDS, let n and $n^{\prime }$ be nodes in (N,E) that have a white-hat bot w and $w^{\prime }$, respectively.*

*w can exchange messages with the BDS if n is observable and controllable by the BDS.*

*$w^{\prime }$ can exchange messages with w if $n^{\prime }$ is observable and controllable by w.*



A message received from a certain white-hat bot can be transferred to another white-hat bot. This message transfer allows a white-hat bot to exchange messages with another white-hat bot that is at a node unobservable and uncontrollable by itself. We define a relation between white-hat bots to express that they can exchange messages through the message transfer.

**Definition** **4** (Link relation between white-hat bots).
*In a network defended by the BDS, let w and $w^{\prime }$ be white-hat bots on the network. w is said to be linked with $w^{\prime }$ if w can exchange messages with $w^{\prime }$ through the message transfer via the white-hat botnet.*


We will depict a link relation between linked white-hat bots as a red dashed line (---). We give a necessary and sufficient condition to determine if two white-hat worms are linked.

**Property** **1.**
*In a network (N,E) defended by the BDS, let n and $n^{\prime }$ be nodes in (N,E) that have a white-hat bot w and $w^{\prime }$, respectively. w is linked with $w^{\prime }$ if and only if there is a path ρ between n and $n^{\prime }$ such that every node on ρ has a white-hat bot (see Figure 5).*


**Proof.** See Section A.1. □

This message transfer mechanism enables the BDS to expand the scope that the BDS can observe and control. The observability and controllability by the BDS via the white-hat botnet is defined as follows.

**Definition** **5** (Observability and controllability by BDS via white-hat botnet).
*In a network defended by the BDS:*

*Let u be a node unobservable by the BDS. u is said to be observable by the BDS via the white-hat botnet if the BDS can check whether there is any bot at u via the white-hat botnet.*

*Let u be a node uncontrollable by the BDS. u is said to be controllable by the BDS via the white-hat botnet if the BDS can send a white-hat worm to u via the white-hat botnet.*



We give a necessary and sufficient condition to determine if a given node unobservable (or uncontrollable) by the BDS is observable (or controllable) by the BDS via the white-hat botnet.

**Property** **2.**
*Let (N,E) be a network defended by the BDS (see Figure 6).*
*(i)* 
*Let u be a node of (N,E) unobservable by the BDS. u is observable by the BDS via the white-hat botnet if and only if there is a white-hat bot w by which u is observable and a white-hat bot $w^{\prime }$ at a node $n^{\prime }$ observable by the BDS such that w is linked with $w^{\prime }$.*
*(ii)* 
*Let u be a node of (N,E) uncontrollable by the BDS. u is controllable by the BDS via the white-hat botnet if and only if there is a white-hat bot w by which u is controllable and a white-hat bot $w^{\prime }$ at a node $n^{\prime }$ controllable by the BDS such that w is linked with $w^{\prime }$.*



**Proof.** See Section A.2. □

This property can be reformulated with the structural characterization of Property 1 as follows.

**Corollary** **1.**
*Let (N,E) be a network defended by the BDS (see Figure 6).*
*(i)* 
*Let u be a node of (N,E) unobservable by the BDS. u is observable by the BDS via the white-hat botnet if and only if there is a path ρ between u and a node $n^{\prime }$ observable by the BDS such that every node on ρ has a white-hat bot except u.*
*(ii)* 
*Let u be a node of (N,E) uncontrollable by the BDS. u is controllable by the BDS via the white-hat botnet if and only if there is a path ρ between u and a node $n^{\prime }$ uncontrollable by the BDS such that every node on ρ has a white-hat bot except u.*



**Proof.** See Section A.3. □

Let us consider Figure 4c. In this state, the white-hat botnet spread even to $n_{14},n_{15},n_{18}$ and $n_{20}$ unobservable and uncontrollable by the BDS and drove out all the malicious bots except at $n_{24}$. The white-hat bots can be divided into three groups by message link depicted by red dashed lines.

The first group is the bots at $n_{1}$, $n_{2}$, $n_{3}$, $n_{4}$ and $n_{7}$ (see the right upper part of Figure 4c). We can confirm that those bots are linked to each other by using Property 1 because for any pair of those nodes, there is a path between them such that every node on the path has a white-hat bot. The BDS can directly command and control all the bots of this group because those bots are at nodes observable and controllable by the BDS.

The second group is the bots at $n_{10}$, $n_{14}$, $n_{15}$ and $n_{18}$ (see the right middle part of Figure 4c). Only $n_{10}$ is observable and controllable by the BDS, the others are not. The BDS can only directly command and control the bot at $n_{10}$ but not others. However, the bots at $n_{10}$ are linked with the other bots. From Property 2, $n_{14}$, $n_{15}$, $n_{18}$ and their adjacent nodes, which are represented by a yellow dot pattern in the figure, are observable and controllable by the BDS via the white-hat botnet. This can be confirmed by using Corollary 1 because for any *u* of those nodes, there is a path between *u* and $n_{10}$ such that every intermediate node on the path has a white-hat bot. This means that the BDS can indirectly command and control the bots of $n_{14}$, $n_{15}$ and $n_{18}$.

The last group is the bot at $n_{20}$ (see the right lower part of Figure 4c). $n_{20}$ is unobservable and uncontrollable by the BDS. Furthermore, since this bot is not linked with the other white-hat bots, $n_{20}$ is not observable and controllable by the BDS via the white-hat botnet from Property 2. Therefore, the BDS cannot command and control the bot. As the result, although this bot discovered a malicious bot at $n_{24}$, the BDS could not know of its existence.

### 4.4. Basic Command and Control Strategy

A white-hat botnet autonomously spreads and therefore requires strategical command and control by the BDS to achieve a given goal.

Different goals require different strategies. The white-hat botnet is known as a double-edged sword. It defends the IoT system against malicious botnets but itself consumes the system’s resources. Therefore, it should not stay at the system after disinfecting the malicious botnet.

We set this as a goal and propose a strategy to achieve the goal as follows.

**Strategy** **1**(Withdrawal Strategy). *Remove the white-hat botnet immediately by the BDS itself.*

This strategy will be applied when the white-hat botnet has disinfected the malicious botnet. In addition, it may be applied when

The white-hat botnet is bloated and has too much waste compared to the benefits; orThe white-hat botnet is too weak to disinfect the malicious botnet and should be rebuilt.

Let us consider Figure 4c again. In this state, no malicious bot remains any longer in the area observable by the BDS directly and indirectly. That is, as far as the BDS knows, the white-hat botnet disinfected all the malicious bots. The BDS will command self-destruction to the white-hat bots according to the Withdrawal strategy. This command reached the white-hat bots not only at controllable nodes $n_{1},n_{2},n_{3},n_{4},n_{7},n_{10}$, but also at uncontrollable nodes $n_{14},n_{15}$ and $n_{18}$ linked with the bot at $n_{10}$. Figure 4d shows the state after those bots self-destructed. Note that the white-hat bot at $n_{20}$ remains because it has not received the self-destruction command. As shown in this example, the BDS can be expected to extend its command and control capability into areas were it cannot observe or control by using the white-hat botnet.

## 5. Simulation Evaluation

In this section, we present the quantitative evaluation of the observability, controllability and basic C&C strategy proposed in the previous section. Molesky states in Reference [25] that the use of white-hat worm technology can be legally enacted by including explicit terms within the Terms and Conditions agreement at the time of purchase. This suggests that white hat worm technology is applicable in real situations. We have also implemented a prototype of BDS that defends 24 Raspberry Pi Zero units and have confirmed that our approach works in a real-world environment. However, because of the complex behavior of BDS, we take a simulation-based approach to its evaluation in this paper.

We first model the behavior of the BDS with an agent-oriented Petri net, called Petri-Nets in a Petri Net (PN^2^). The PN^2^ model enables us to observe the evolution of the BDS over time. We perform two experiments to evaluate the proposal from different perspectives. One is a special case that has hierarchical observability and controllability based on the network structure. The other is a general case in which observability and controllability are decided at random.

### 5.1. Modeling and Simulation

In the evaluation, a key behavior of the BDS is the battle between malicious and white-hat botnets. The battle can be regarded as a multi-agent system, i.e., the devices (these may be bots) and the network, respectively, correspond to agents and an environment that the agents act on. PN^2^ is a mathematical and graphical modeling tool to study multi-agent systems. A PN^2^ consists of two levels of Petri nets: agent-nets and an environment net. An agent net represents the behavior of an agent. The environment net contains those agent nets as tokens and represents the interaction among the agents. For the details of modeling, refer to Reference [23]. The feature of PN^2^ is that its model is a mathematical description as well as a graphical representation and is also executable. The last two features help to make the behavior of the model easier to understand. Playing the key role are tokens. With tokens, the PN^2^ model serves as a “game board” and the system behavior becomes “token games”.

We have developed a tool called PN2Simulator that can edit, simulate and analyze PN^2^ and we use it to study the behavior of BDS interactively.

Figure 7 shows a screenshot of the PN2Simulator, which displays the PN^2^ model representing the state shown in Figure 4b. The left side of the screen represents the environmental net, while the right side represents the agent nets. The environment net is an extension of Petri nets. It consists of two types of nodes called places and transitions. Places are drawn as circles (◯) and transitions are drawn as boxes (□). Each place represents a network node. The 4 × 6 places in total are arranged as 4 × 6 in the same way as the nodes in Figure 4b. Places can hold tokens. Tokens are drawn as ellipses (⬭). Each token represents an agent such as an IoT device, a malicious worm or a white-hat worm and is labeled with the corresponding agent’s name.

Let us consider place P1 (top left). P1 holds two tokens representing a device and a malicious worm. This indicates that the device was infected by the worm and has been a malicious bot. Next, let us consider place P5 (below P1). P5 holds only one token representing the device. This means that the device is not infected by any worm and is normal. Then, let us consider place P10 (to the lower right of P5). P10 stores two tokens representing a device and a white-hat worm launched by the BDS. This means that the device was infected by the white-hat worm and has been a white-hat bot.

Each transition represents an interaction between agents. Recall that each token corresponds to an agent net. A transition in an environment net fires synchronously with transitions in one or more agent nets. This synchronization represents an interaction between the corresponding agents. The conditions for firing are described by labels, not transition identifiers. Therefore, the agent nets involved in firing are determined by dynamic coupling by labels. The PN2Simulator highlights transitions that can fire in red. Transition T123 is fireable. This indicates that the malicious worm at P1 is trying to infect the device at P5. A firing of this transition results in a copy of the malicious worm at P5. Then the copy turns the device at P5 into a malicious bot. For more information on the firing rule, refer to Ref. [23].

### 5.2. Evaluation Indicators

We use the following two indicators in the evaluation.

Infection rate of malicious bots at simulation step *t*:
$$R_{mal}\left(t\right)\overset{def}{=}\frac{{\#}_{mal}\left(t\right)}{{\#}_{node}}=\frac{{\#}_{mal}^{obs}\left(t\right)+{\#}_{mal}^{in-obs}\left(t\right)+{\#}_{mal}^{unobs}\left(t\right)}{{\#}_{node}},$$
where–${\#}_{node}$ is the number of nodes, i.e., devices, in the given network;–${\#}_{mal}\left(t\right)$ is the number of malicious bots in the whole network at *t*. This breaks down into:${\#}_{mal}^{obs}\left(t\right)$ is the number of malicious bots on the nodes observable by the BDS at *t*;${\#}_{mal}^{in-obs}\left(t\right)$ is the number of malicious bots on the nodes indirectly observable by the BDS via the white-hat botnet at *t*;${\#}_{mal}^{unobs}\left(t\right)$ is the number of malicious bots on the nodes unobservable by the BDS even via the white-hat botnet at *t*.Infection rate of white-hat bots at simulation step *t*:
$$R_{wh}\left(t\right)\overset{def}{=}\frac{{\#}_{wh}\left(t\right)}{{\#}_{node}}=\frac{{\#}_{wh}^{obs}\left(t\right)+{\#}_{wh}^{in-obs}\left(t\right)+{\#}_{wh}^{unobs}\left(t\right)}{{\#}_{node}},$$
where–${\#}_{wh}\left(t\right)$ is the number of white-hat bots in the whole network at *t*. This breaks down into:${\#}_{wh}^{obs}\left(t\right)$ is the number of white-hat bots on the nodes observable by the BDS at *t*;${\#}_{wh}^{in-obs}\left(t\right)$ is the number of white-hat bots on the nodes indirectly observable by the BDS via the white-hat botnet at *t*;${\#}_{wh}^{unobs}\left(t\right)$ is the number of white-hat bots on the nodes unobservable by the BDS even via the white-hat botnet at *t*.

The smaller the values of both $R_{mal}$ and $R_{wh}$ the better. In particular, $R_{mal}$ should be zero, which means the malicious botnet is wiped out.

### 5.3. Hierarchical Case

We first target a case in which observability and controllability are given based on the network structure. Figure 8 illustrates the network. This network has a two-level hierarchy and consists of one upper-level network and four lower-level networks. The upper-level network consists of 36 nodes that are both observable and controllable by the BDS, while each lower-level network consists of 16 nodes that are neither unobservable nor uncontrollable by the BDS. Therefore, the number ${\#}_{node}^{obs}$ of nodes observable and controllable by the BDS in the whole network is 36. Each lower-level network is connected to only the upper-level network by a single connection. This single connection is intended to reduce the potential for the spread of infection between networks.

The given network can be systematically translated into a PN^2^ model. Intuitively, each node of the network corresponds to a single place in the environment net of the PN^2^ model. A single node may have multiple connections. A single connection is represented by the same colored parts of the net. If there is no certain connection, we would simply omit the corresponding part. For example, let us consider the 37th node. It has four connections leading to the 27th, 36th, 38th and 47th nodes. Therefore, those connections are represented by using all the colored parts of Figure 8. On the other hand, let us consider the 27th node. It has only three connections. Therefore, the corresponding red part is omitted.

A single trial of simulation follows the steps below.

Let t=0 when the BDS detects a malicious botnet and launches white-hat worms to build a white-hat botnet. Let $0^{-}$ and $0^{+}$, respectively, denote before and after launching the white-hat worms. Assume the malicious botnet consists of 20 bots, i.e., ${\#}_{mal}\left(0^{-}\right)=20$ and they are placed at random in the whole network.The BDS adopts the few-elite strategy and randomly sends 10 white-hat worms to observable nodes, i.e., ${\#}_{wh}\left(0^{+}\right)={\#}_{wh}^{obs}\left(0^{+}\right)=10$. Figure 9a shows an example of the state at $0^{+}$. Assume the white-hat worms have the capabilities below.Lifespan *ℓ* is 3, 4 or 5 stepsPossibility of secondary infection ρ is 50%Assume each device reboots at 11 steps after infection.When t=10,000, the BDS adopts the Withdrawal strategy and commands the white-hat botnet to self-destruct. Let ${10,000}^{-}$ and ${10,000}^{+}$, respectively, denote before and after sending the self-destruction order. Figure 9b shows an example of the state at ${10,000}^{-}$.

We ran 1000 trials of the simulation by changing the lifespan *ℓ* of white-hat worms as 3, 4 and 5 and then calculated the mean of each measure and indicator.

Table 2 shows the result. Table 2a is for ℓ=3. The values of each row, respectively, show the mean of ${\#}_{mal}^{obs}\left(t\right)$, ${\#}_{mal}^{in-obs}\left(t\right)$, ${\#}_{mal}^{unobs}\left(t\right)$, $R_{mal}\left(t\right)$, ${\#}_{wh}^{obs}\left(t\right)$, ${\#}_{wh}^{in-obs}\left(t\right)$, ${\#}_{wh}^{unobs}\left(t\right)$ and $R_{wh}\left(t\right)$ for 1000 trials. When $t=0^{+}$, the BDS built a white-hat botnet composed of 10 bots on the observable network. This white-hat botnet increased the number of malicious bots recognized by the BDS from 5.79 ($=\bar{{\#}_{mal}^{obs}\left(0^{-}\right)}$) to 6.05 ($=\bar{{\#}_{mal}^{obs}\left(0^{+}\right)}+\bar{{\#}_{mal}^{obs}\left(0^{+}\right)}$). Then, when $t={10,000}^{+}$, the BDS commanded the white-hat botnet to self-destruct. This decreased the number of white-hat bots from 16.8 ($=\bar{{\#}_{wh}^{obs}\left({10,000}^{-}\right)}+\bar{{\#}_{wh}^{in-obs}\left({10,000}^{-}\right)}+\bar{{\#}_{wh}^{unobs}\left({10,000}^{-}\right)}$) to 3.08 ($=\bar{{\#}_{wh}^{unobs}\left({10,000}^{+}\right)}$. Similarly, Table 2b,c are, respectively, for ℓ=4 and 5.
Figure 9A state transition of the BDS for the IoT network with hierarchical observability and controllability. (**a**) State at t=0+. The BDS sent white-hat worms to 10 observable nodes in order to disinfect 20 malicious bots; (**b**) State at t=10,000−. The BDS will command self-destruction to the white-hat botnet according to the Withdrawal strategy.
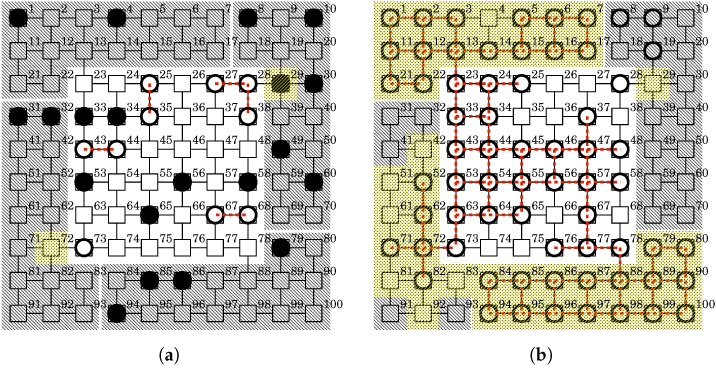


Figure 10 illustrates the result. Figure 10a is for $t={10,000}^{-}$. The horizontal axis shows *ℓ* and the vertical axis shows the mean of bots. It is a stacked bar graph. The red bar represents the number of malicious bots and can be divided into $\bar{{\#}_{mal}^{obs}\left({10,000}^{-}\right)}$, $\bar{{\#}_{mal}^{in-obs}\left({10,000}^{-}\right)}$ and $\bar{{\#}_{mal}^{unobs}\left({10,000}^{-}\right)}$. The blue bar represents the number of white-hat bots and can be divided in the same way as the red bar.

When ℓ=3, the malicious bots increased from 20.00 ($=\bar{{\#}_{mal}}\left(0^{-}\right)$) to 67.39 ($=\bar{{\#}_{mal}}\left({10,000}^{-}\right)$). This is because the white-hat worm with a short lifespan died soon and therefore is difficult to spread in the hierarchical network. Figure 10b shows the state when $t={10,000}^{+}$. Even if the BDS commanded the white-hat botnet to self-destruct, 67.39 malicious bots and 3.08 white-hat bots (total 70.47 bots) remained. On the other hand, when ℓ=4, the malicious bots decreased from 20.00 to 2.81. This means the white-hat botnet with such a long lifespan is useful to disinfect the malicious botnet. However, the white-hat bots increased from 10.00 ($=\bar{{\#}_{wh}}\left(0^{+}\right)$) to 73.28 ($=\bar{{\#}_{wh}}\left({10,000}^{-}\right)$). After that, the BDS commanded the white-hat botnet to self-destruct and removed 62.87 ($=\bar{{\#}_{wh}^{obs}\left({10,000}^{-}\right)}+\bar{{\#}_{wh}^{in-obs}\left({10,000}^{-}\right)}$) white-hat bots. This means the BDS removed 85.8% of the white-hat bots, including those in the unobservable networks. Furthermore, when ℓ=5, the BDS wiped out all the malicious bots and then removed 76.34 white-hat bots. This means 90.32% of white-hat bots were removed.

Let us evaluate the two indicators, the infection rate of malicious bots $R_{mal}$ and the infection rate of white-hat bots $R_{wh}$. The BDS reduced $R_{mal}$ from 20% ($=R_{mal}\left(0^{-}\right)$) to 0% ($=R_{mal}\left({10,000}^{+}\right)$) and also reduced $R_{wh}$ from 10% ($=R_{wh}\left(0^{+}\right)$) to 8.18% ($=R_{wh}\left({10,000}^{+}\right)$) if ℓ=5. Almost the same trend occurred for ℓ=4. These results show that even in the case in which observability and controllability are given hierarchically, the Withdrawal strategy works effectively if the white-hat worm has enough infectivity (ℓ≥4 in this example).

### 5.4. General Case

We consider another case in which observability and controllability are given regardless of the network structure. This can be regarded as a general case. The network has a square lattice topology and consists of 10 × 10 nodes. Given the total number of observable and controllable nodes, we decide the observability and controllability for each node at random. The given network can be systematically translated into a PN^2^ model in the same way as the previous case. Figure 11 shows an example of state-transition in this case. Figure 11a shows an example of the state at $0^{+}$. In this example, 30 nodes are both observable and controllable. The malicious botnet infected 20 nodes at random regardless of observability and controllability. The white-hat worms were launched at 10 of 30 observable and controllable nodes such as the 21st and 24th nodes. Figure 11b shows an example of the state at ${10,000}^{-}$. The BDS disinfected the malicious botnet but has left 26 white-hat bots on the network. The BDS will command self-destruction to the white-hat botnet according to the Withdrawal strategy. This results in 24 of 26 bots being removed.

A single trial of simulation follows the same as the previous case. However, we added the following parameters in addition to lifespan for generality.

Number ${\#}_{node}^{obs}$ of observable and controllable nodes is 10, 20 or 30Number ${\#}_{wh}\left(0^{+}\right)$ of white-hat worms launched by the BDS is 10, 20 or 30, but ${\#}_{wh}\left(0^{+}\right)\leq {\#}_{node}^{obs}$.

Table 3 shows the result. Table 3a is for ℓ=3. The rows are bundled by ${\#}_{node}^{obs}$. The values of each row, respectively, show ${\#}_{node}^{obs}$, $\bar{{\#}_{wh}^{obs}\left(0^{+}\right)}$, $\bar{{\#}_{mal}^{obs}\left({10,000}^{-}\right)}$, $\bar{{\#}_{mal}^{in-obs}\left({10,000}^{-}\right)}$, $\bar{{\#}_{mal}^{unobs}\left({10,000}^{-}\right)}$, $\bar{R_{mal}\left({10,000}^{-}\right)}$, $\bar{R_{mal}\left({10,000}^{+}\right)}$, $\bar{{\#}_{wh}^{obs}\left({10,000}^{-}\right)}$, $\bar{{\#}_{wh}^{in-obs}\left({10,000}^{-}\right)}$, $\bar{{\#}_{wh}^{unobs}\left({10,000}^{-}\right)}$, $\bar{R_{wh}\left({10,000}^{-}\right)}$ and $\bar{R_{wh}\left({10,000}^{+}\right)}$ for 1000 trials. Let us first consider the case for ${\#}_{node}^{obs}=10$ and $\bar{{\#}_{wh}^{obs}\left(0^{+}\right)}=10$. When $t=0^{+}$, the BDS sent white-hat worms to all the observable and controllable nodes and built a white-hat botnet composed of 10 bots. When $t={10,000}^{-}$, the white-hat botnet reduced the number of malicious bots from 20.0 to 5.04. Similar to the hierarchical case, the white-hat botnet with a short lifespan could not wipe out the malicious botnet. Then, when $t={10,000}^{+}$, the BDS commanded the white-hat botnet to self-destruct. This drastically decreased the number of white-hat bots from 72.47 to 1.21 even though the observable and controllable bots numbered just 7.28. The BDS removed 98.3% (=(72.47−1.21)/74.27) of the white-hat bots. Next, let us consider the case for ${\#}_{node}^{obs}\geq 20$ and $\bar{{\#}_{wh}^{obs}\left(0^{+}\right)}=10$. The BDS removed more white-hat bots and reduced the number of remaining bots to 0.77 when ${\#}_{node}^{obs}=30$.

Table 3b is for ℓ=4. Unlike in the case of ℓ=3, the BDS could wipe out the malicious botnet with the white-hat botnet by $t={10,000}^{-}$. When $t={10,000}^{+}$, the BDS commanded the white-hat botnet to self-destruct and reduced the number of white-hat bots from 87.19 to 0.53 even when ${\#}_{node}^{obs}=10$ and $\bar{{\#}_{wh}^{obs}\left(0^{+}\right)}=10$. Similarly, Table 3c is for ℓ=5. It indicates a similar tendency.

Figure 12a–c are, respectively, stacked bar charts that illustrate Table 3a–c. Figure 12a is for ℓ=3. The horizontal axis shows ${\#}_{node}^{obs}$. The vertical axis shows the mean of bots. Each bar represents the same meaning as in Figure 10. As mentioned before, although the white-hat botnet fought the malicious botnet well, several malicious bots remained. They remained regardless of ${\#}_{node}^{obs}$. This is because the white-hat botnet autonomously spreads regardless of the BDS’s observability and controllability. On the other hand, $\bar{{\#}_{mal}^{obs}\left({10,000}^{-}\right)}$ and $\bar{{\#}_{wh}^{obs}\left({10,000}^{-}\right)}$ increased with ${\#}_{node}^{obs}$. This is because the more ${\#}_{node}^{obs}$, the more the number of bots observable by the BDS. More than 98% of the white-hat bots were removed regardless of ${\#}_{node}^{obs}$. Figure 12b,c are for ℓ=4 and 5. As stated before, the white-hat botnet wiped out the malicious botnet. $\bar{{\#}_{wh}^{obs}\left({10,000}^{-}\right)}$ increased with ${\#}_{node}^{obs}$ for the same reason.

Let us evaluate the two indicators $R_{mal}$ and $R_{wh}$ in the general case. If ℓ≥4, the BDS reduced $R_{mal}$ from 20% ($=R_{mal}\left(0^{-}\right)$) to 0% ($=R_{mal}\left({10,000}^{+}\right)$) and also reduced $R_{wh}$ from 10% ($=R_{wh}\left(0^{+}\right)$) to less than 1% ($=R_{wh}\left({10,000}^{+}\right)$). Even if ℓ=3, the values of both indicators were reduced from the initial. These results show that even in the case in which observability and controllability are given regardless of the network structure, the Withdrawal strategy works well in the general case, as long as the white-hat worms are sufficiently infectious.

### 5.5. Discussion

We show the positioning of this study within the relevant studies in Table 4. Most of the related studies deal with the detection, mitigation and spread prevention techniques of malicious botnets. As mentioned in Section 2, these existing techniques are useful in mitigating botnet threats. However, they did not intend to disinfect botnets that have already spread. In contrast, the last three references aim to disinfect malicious botnets. Reference [23] proposed the concept and the system design of BDS to use white-hat botnets to disinfect malicious botnets. It also proposed a basic launcher and strategies to build the white-hat botnet. Reference [26] proposed an improved launcher adopting machine learning. The launcher learns how to build white-hat botnets appropriate to the infection status of the malicious botnet and can disinfect given botnets effectively. Thus, building white-hat botnets has been studied well. On the other hand, command and control of built botnets have not been much discussed.

In this paper, we introduced the concept of observability and controllability to enable the consideration of communications between BDS and botnets over the Internet. Observability and controllability over botnets have not been discussed at all so far. We proved that BDS can extend the observability and controllability by means of the white-hat bots’ message transmission. In addition, we proposed the Withdrawal strategy as a basic C&C strategy. No C&C strategy has been proposed so far. The Withdrawal strategy is a simple one, but it demonstrates the potential of actively controlling white-hat botnets to further enhance the defense capabilities.

IoT systems are open systems. As nodes can be dynamically added or removed, BDS may not be able to observe or control all nodes. This means that observability and controllability are limited and may create cyber security risks in botnet detection and command and control. Combining BDS with white-hat botnets can reduce this limitation. BDS can use self-propagating botnets to indirectly observe and control nodes that BDS cannot directly observe or control through the message link over the botnet. As a result, BDS can detect and remove botnets even if the network configuration of an IoT system changes.

The BDS intentionally infects IoT systems with white-hat botnets. This is like a vaccine. The white-hat botnets have the side effect that they can also infect normal devices and consume their resources. However, due to the white-hat worms’ lifespan and the Withdrawal strategy, the infection is temporary and has limited impact. The BDS itself requires little cost to introduce and operate. On the other hand, research and development of excellent white-hat botnets, building strategies and C&C strategies may be costly. By opening up research and development, we expect not only to reduce the cost needed but also to produce innovative results.

## 6. Conclusions

In this paper, we proposed a mechanism to command and control the self-built white-hat botnet for disinfecting malicious botnets in the network including unobservable and uncontrollable nodes. We first introduced the concept of observability and controllability into the BDS. Next, we theoretically clarified the mechanism to command and control the botnet under observability and controllability. In addition, we proposed a strategy to command and control the botnet reasonably as a Withdrawal strategy. Then, we modeled the mechanism and the strategy with agent-oriented Petri net PN^2^ and confirmed the effect of the mechanism and the strategy through the simulation of the model. The result shows that even if considering observability and controllability, the BDS wiped out the malicious bots and reduced the white-hat bots to less than 1% as long as the white-hat worms are sufficiently infectious.

We are currently working on the following challenges.

(i)Resident strategy of white-hat botnets: Malicious botnets will invade IoT systems not just once, but many times. This strategy aims to make a small number of the white-hat bots stay permanently so as to promptly respond to invading botnets at an earlier stage.(ii)Heterogeneous strategy of white-hat botnets: Like Mirai, white hat botnets have variants with different capabilities. This strategy aims to synergically enhance defense by operating multiple variants in combination.(iii)Reinforcement learning-based planning of building and C&C strategies of white-hat botnets: Supervised learning-based planning of strategies has been proposed in Reference [27] but requires data collection and labeling before learning. Reinforcement learning works well with multi-agent systems and is expected to dynamically optimize strategies without human intervention.(iv)Not only technical but also ethical, legal and social issues (ELSI), empirical testing and system implementation.

These challenges will provide new impetus and make this research area even more exciting.

## Figures and Tables

**Figure 1 sensors-22-09423-f001:**
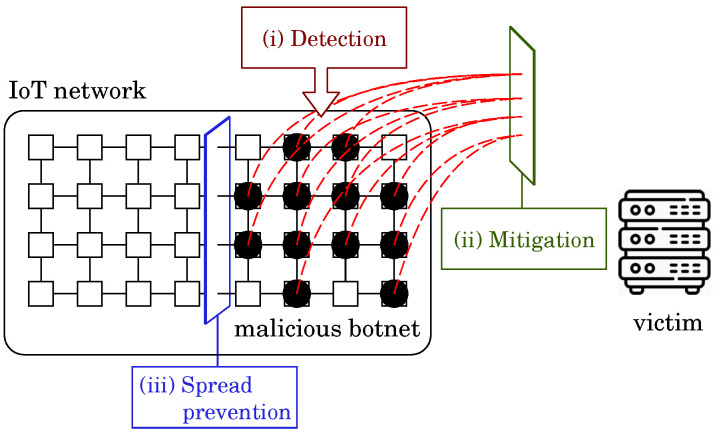
Countermeasure techniques against malicious botnet threats.

**Figure 2 sensors-22-09423-f002:**
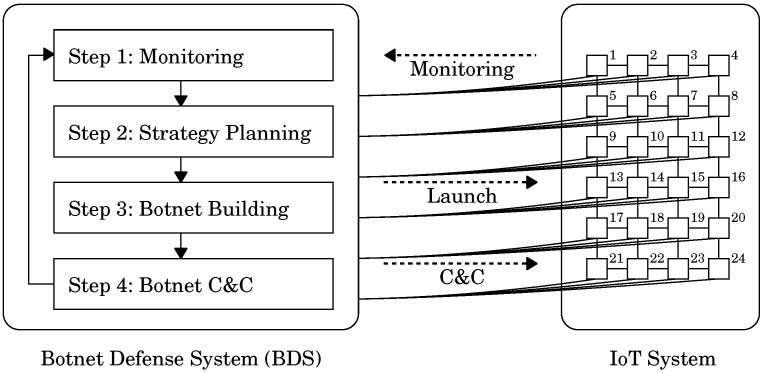
System configuration of BDS.

**Figure 3 sensors-22-09423-f003:**
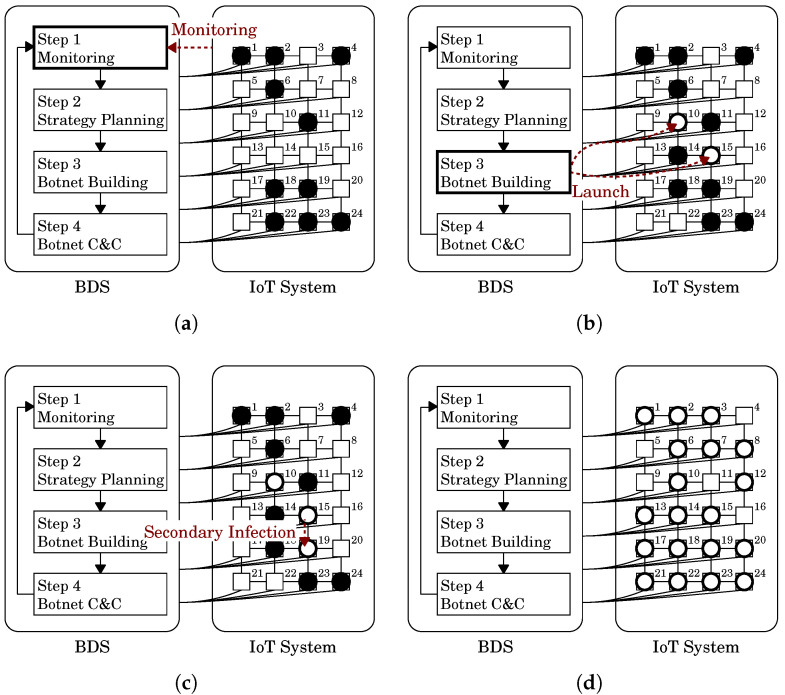
A state transition of the BDS, which illustrates how the BDS works. (**a**) State when the BDS detected a malicious botnet, which consists of 10 bots (●). The BDS decided to adopt the few-elite strategy [23] for disinfecting the malicious botnet; (**b**) State in botnet building of Step 3. The BDS sent white-hat worms to the 10th and 15th nodes and built a white-hat botnet composed of two bots (○); (**c**) State when the white-hat botnet spread by infecting the 19th device that had already been a malicious bot; (**d**) State in which the white-hat botnet wiped out the malicious botnet. However, many white-hat bots still stay on.

**Figure 4 sensors-22-09423-f004:**
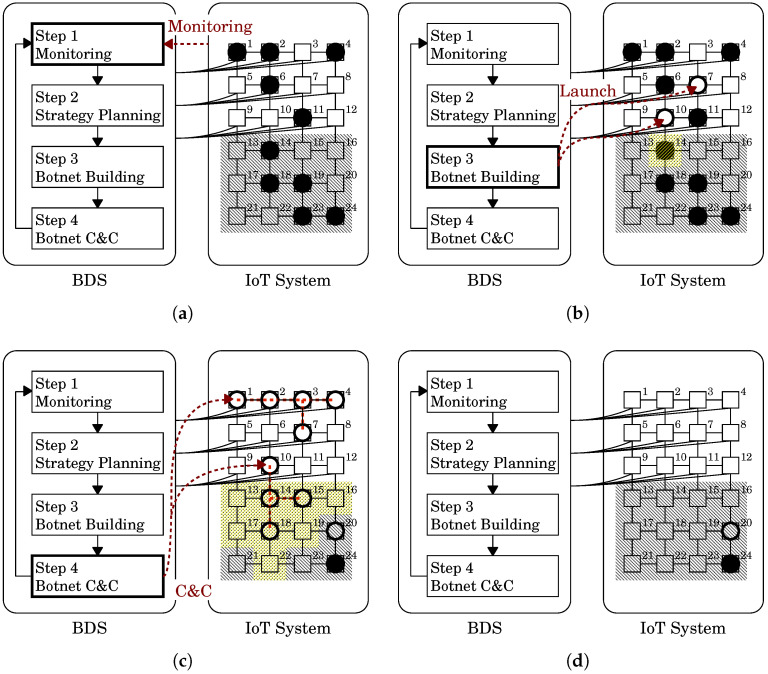
A state transition of the BDS for the IoT system with restricted observability and controllability. (**a**) State when the BDS detected a malicious botnet. The unobservable and uncontrollable nodes are shaded. The BDS could check only the observable nodes and found only 5 out of 10 bots; (**b**) State in botnet building of Step 3. The BDS sent two white-hat worms to the controllable nodes, $n_{7}$ and $n_{10}$ and built the initial white-hat botnet. Each white-hat bot can observe and control adjacent nodes instead of the BDS; (**c**) State in which the white-hat botnet spread to even the unobservable and uncontrollable nodes and almost disinfected the malicious botnet. The BDS can indirectly observe and control the nodes shaded in yellow patterns through the message links (---) over the botnet. In Step 4, the BDS commanded self-destruction to the botnet; (**d**) State after the white-hat bots receiving the command self-destroyed. The command reached the white-hat bots not only at controllable nodes $n_{1},n_{2},n_{3},n_{4},n_{7},n_{10}$ but also at uncontrollable nodes $n_{14},n_{15}$ and $n_{18}$ linked with the bot at $n_{10}$. Two bots remained.

**Figure 5 sensors-22-09423-f005:**
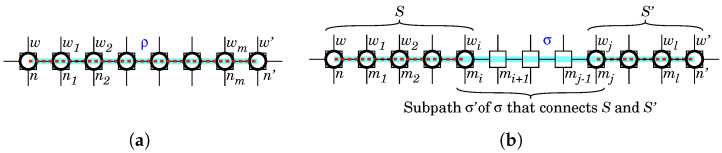
Illustration of Property 1. (**a**) “If” part; (**b**) “Only if” part.

**Figure 6 sensors-22-09423-f006:**
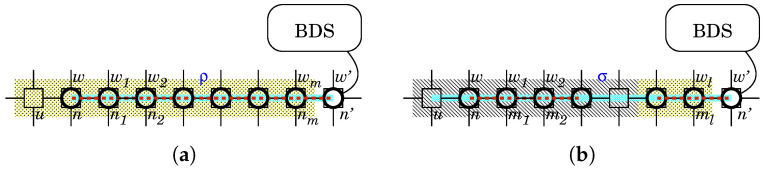
Illustration of Property 2 and Corollary 1. (**a**) “If” part; (**b**) “Only if” part.

**Figure 7 sensors-22-09423-f007:**
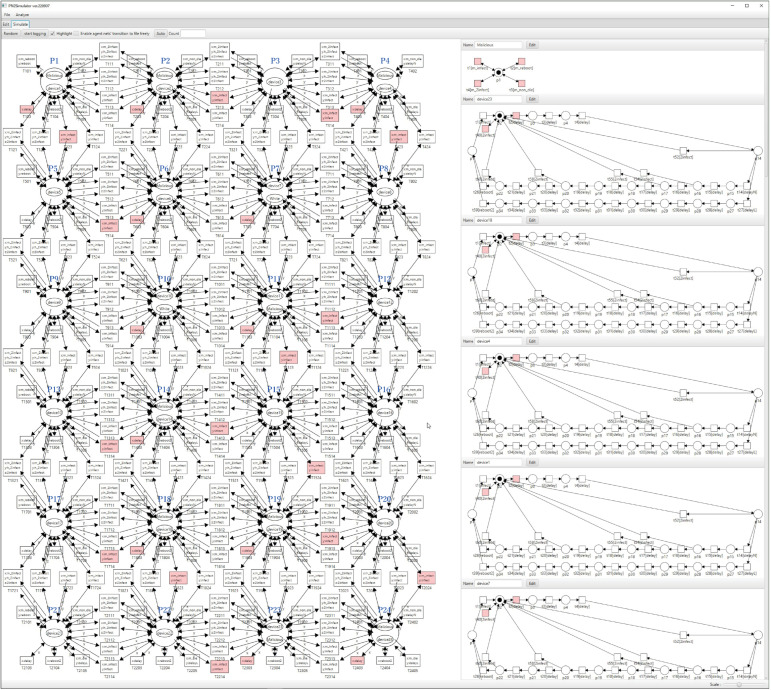
A screenshot of the PN2Simulator, which displays the PN^2^ model representing the state shown in Figure 4b.

**Figure 8 sensors-22-09423-f008:**
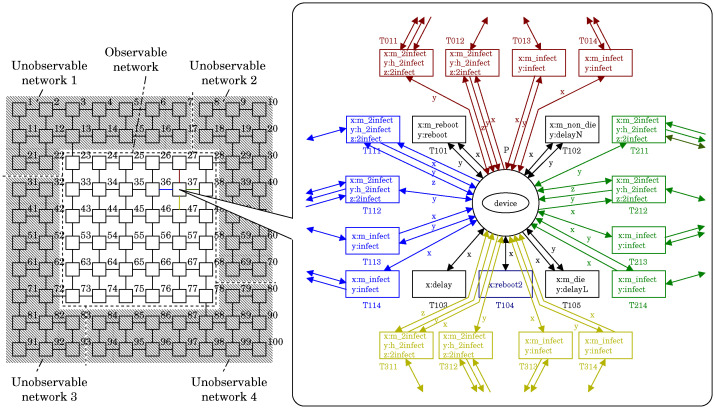
The network used in the hierarchical case.

**Figure 10 sensors-22-09423-f010:**
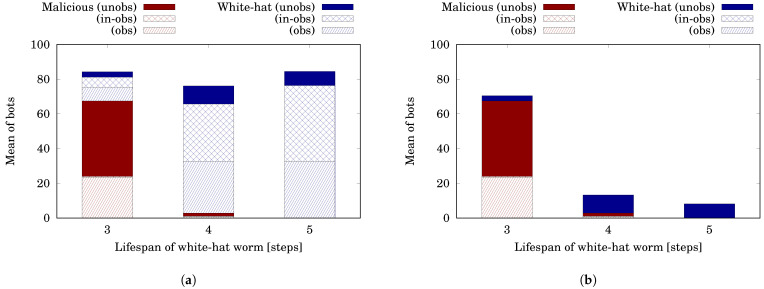
Simulation results in the hierarchical case. (**a**) Before Withdrawal (${10,000}^{-}$ steps); (**b**) After Withdrawal (${10,000}^{+}$ steps).

**Figure 11 sensors-22-09423-f011:**
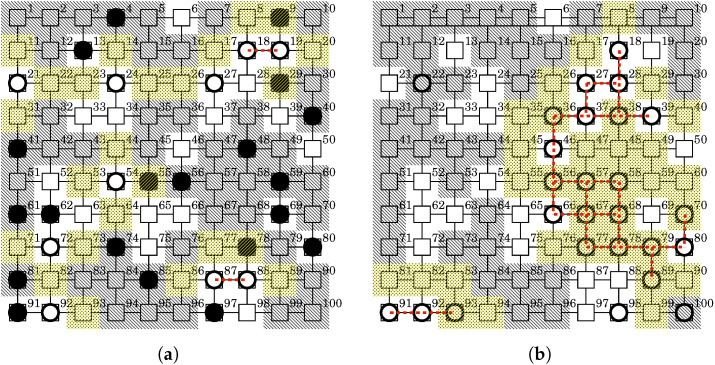
A state transition of the BDS for an IoT network with randomly determined observability and controllability. (**a**) State at $t=0^{+}$. The BDS sent white-hat worms to 10 of 30 observable nodes in order to disinfect 20 malicious bots; (**b**) State at $t={10,000}^{-}$. The BDS will command self-destruction to the white-hat botnet according to the Withdrawal strategy.

**Figure 12 sensors-22-09423-f012:**
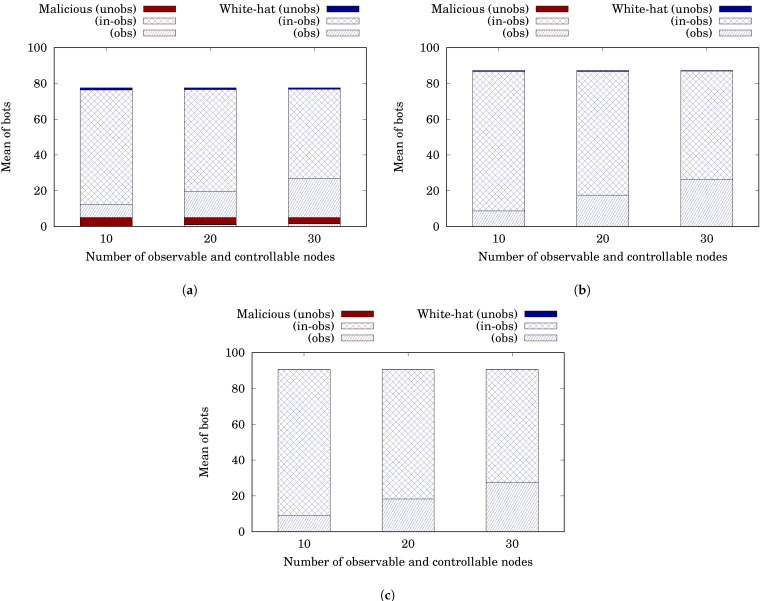
Simulation results in the general case. (**a**) Lifespan ℓ=3 steps; (**b**) Lifespan ℓ=4 steps; (**c**) Lifespan ℓ=5 steps.

**Table 1 sensors-22-09423-t001:** Countermeasure techniques.

Category	Techniques
Detection	Signature-based (Said et al. [9])
	Anomaly and host-based (Bezerra et al. [10], Soe et al. [11])
	Anomaly and network-based (Meidan et al. [12], Hoang et al. [13])
	Looking-Back Approach (Mihoub et al. [14])
Mitigation	Software-defined network control (Manso et al. [15])
	Adaptive network layer (Ceron et al. [16])
	Open source extensible tool (Khattak et al. [17])
Spread prevention	Mitigation and preventive steps (US-CERT [3])
	Hardening and prevention scripts (Frank et al. [18])
	Whitelist-based solution (Gopal et al. [19])

**Table 2 sensors-22-09423-t002:** Simulation results in the hierarchical case.

(a) Lifespan ℓ=3 Steps								
**Step** t	**Malicious Bots**				**White-Hat Bots**			
	$\bar{{\#}_{\mathrm{mal}}^{\mathrm{obs}}}$	$\bar{{\#}_{\mathrm{mal}}^{\mathrm{in}-\mathrm{obs}}}$	$\bar{{\#}_{\mathrm{mal}}^{\mathrm{unobs}}}$	$\bar{R_{\mathrm{mal}}}$	$\bar{{\#}_{\mathrm{wh}}^{\mathrm{obs}}}$	$\bar{{\#}_{\mathrm{wh}}^{\mathrm{in}-\mathrm{obs}}}$	$\bar{{\#}_{\mathrm{wh}}^{\mathrm{unobs}}}$	$\bar{R_{\mathrm{wh}}}$
$0^{-}$	5.79	0.00	14.21	20.00%	0.00	0.00	0.00	0.00%
$0^{+}$	5.79	0.26	13.95	20.00%	10.00	0.00	0.00	10.00%
${10,000}^{-}$	23.48	0.51	43.40	67.39%	7.55	6.17	3.08	16.80%
${10,000}^{+}$	23.48	0.51	43.40	67.39%	0.00	0.00	3.08	3.08%
**(b) Lifespan** ℓ=4 **Steps**								
**Step** t	**Malicious Bots**				**White-Hat Bots**			
	$\bar{{\#}_{\mathrm{mal}}^{\mathrm{obs}}}$	$\bar{{\#}_{\mathrm{mal}}^{\mathrm{in}-\mathrm{obs}}}$	$\bar{{\#}_{\mathrm{mal}}^{\mathrm{unobs}}}$	$\bar{R_{\mathrm{mal}}}$	$\bar{{\#}_{\mathrm{wh}}^{\mathrm{obs}}}$	$\bar{{\#}_{\mathrm{wh}}^{\mathrm{in}-\mathrm{obs}}}$	$\bar{{\#}_{\mathrm{wh}}^{\mathrm{unobs}}}$	$\bar{R_{\mathrm{wh}}}$
$0^{-}$	5.79	0.00	14.21	20.00%	0.00	0.00	0.00	0.00%
$0^{+}$	5.79	0.26	13.95	20.00%	10.00	0.00	0.00	10.00%
${10,000}^{-}$	0.71	0.38	1.72	2.81%	29.81	33.06	10.41	73.28%
${10,000}^{+}$	0.71	0.38	1.72	2.81%	0.00	0.00	10.41	10.41%
**(c) Lifespan** ℓ=5 **Steps**								
**Step** t	**Malicious Bots**				**White-Hat Bots**			
	$\bar{{\#}_{\mathrm{mal}}^{\mathrm{obs}}}$	$\bar{{\#}_{\mathrm{mal}}^{\mathrm{in}-\mathrm{obs}}}$	$\bar{{\#}_{\mathrm{mal}}^{\mathrm{unobs}}}$	$\bar{R_{\mathrm{mal}}}$	$\bar{{\#}_{\mathrm{wh}}^{\mathrm{obs}}}$	$\bar{{\#}_{\mathrm{wh}}^{\mathrm{in}-\mathrm{obs}}}$	$\bar{{\#}_{\mathrm{wh}}^{\mathrm{unobs}}}$	$\bar{R_{\mathrm{wh}}}$
$0^{-}$	5.79	0.00	14.21	20.00%	0.00	0.00	0.00	0.00%
$0^{+}$	5.79	0.26	13.95	20.00%	10.00	0.00	0.00	10.00%
${10,000}^{-}$	0.00	0.00	0.00	0.00%	32.72	43.62	8.18	84.52%
${10,000}^{+}$	0.00	0.00	0.00	0.00%	0.00	0.00	8.18	8.18%

**Table 3 sensors-22-09423-t003:** Simulation results in the general case.

(a) Lifespan ℓ=3 Steps											
${\#}_{\mathrm{node}}^{\mathrm{obs}}$	$\bar{{\#}_{\mathrm{wh}}\left(0^{+}\right)}$	**Malicious Bots**					**White-Hat Bots**				
		**Step** $10,000^{-}$				**Step** $10,000^{+}$	**Step** $10,000^{-}$				**Step** $10,000^{+}$
		$\bar{{\#}_{\mathrm{mal}}^{\mathrm{obs}}}$	$\bar{{\#}_{\mathrm{mal}}^{\mathrm{in}-\mathrm{obs}}}$	$\bar{{\#}_{\mathrm{mal}}^{\mathrm{unobs}}}$	$\bar{R_{\mathrm{mal}}}$	$\bar{R_{\mathrm{mal}}}$	$\bar{{\#}_{\mathrm{wh}}^{\mathrm{obs}}}$	$\bar{{\#}_{\mathrm{wh}}^{\mathrm{in}-\mathrm{obs}}}$	$\bar{{\#}_{\mathrm{wh}}^{\mathrm{unobs}}}$	$\bar{R_{\mathrm{wh}}}$	$\bar{R_{\mathrm{wh}}}$
10	10	0.51	0.20	4.33	5.04%	5.04%	7.28	63.99	1.21	72.47%	1.21%
20	10	1.01	0.18	3.84	5.04%	5.04%	14.67	56.83	0.97	72.47%	0.97%
	20	1.49	0.34	5.66	7.49%	7.49%	14.22	55.78	0.83	70.83%	0.83%
30	10	1.52	0.17	3.35	5.04%	5.04%	21.96	49.75	0.77	72.47%	0.77%
	20	2.24	0.30	4.95	7.49%	7.49%	21.42	48.68	0.73	70.83%	0.73%
	30	2.38	0.11	5.45	7.94%	7.94%	20.49	47.33	0.56	68.38%	0.56%
**(b) Lifespan** ℓ=4 **Steps**											
${\#}_{\mathrm{node}}^{\mathrm{obs}}$	$\bar{{\#}_{\mathrm{wh}}\left(0^{+}\right)}$	**Malicious Bots**					**White-Hat Bots**				
		**Step** $10,000^{-}$				**Step** $10,000^{+}$	**Step** $10,000^{-}$				**Step** $10,000^{+}$
		$\bar{{\#}_{\mathrm{mal}}^{\mathrm{obs}}}$	$\bar{{\#}_{\mathrm{mal}}^{\mathrm{in}-\mathrm{obs}}}$	$\bar{{\#}_{\mathrm{mal}}^{\mathrm{unobs}}}$	$\bar{R_{\mathrm{mal}}}$	$\bar{R_{\mathrm{mal}}}$	$\bar{{\#}_{\mathrm{wh}}^{\mathrm{obs}}}$	$\bar{{\#}_{\mathrm{wh}}^{\mathrm{in}-\mathrm{obs}}}$	$\bar{{\#}_{\mathrm{wh}}^{\mathrm{unobs}}}$	$\bar{R_{\mathrm{wh}}}$	$\bar{R_{\mathrm{wh}}}$
10	10	0.00	0.00	0.00	0.00%	0.00%	8.72	77.93	0.53	87.19%	0.53%
20	10	0.00	0.00	0.00	0.00%	0.00%	17.55	69.16	0.48	87.19%	0.48%
	20	0.00	0.00	0.00	0.00%	0.00%	17.38	69.08	0.36	86.83%	0.36%
30	10	0.00	0.00	0.00	0.00%	0.00%	26.30	60.52	0.37	87.19%	0.37%
	20	0.00	0.00	0.00	0.00%	0.00%	26.19	60.31	0.32	86.83%	0.32%
	30	0.00	0.00	0.00	0.00%	0.00%	26.18	60.64	0.29	87.11%	0.29%
**(c) Lifespan** ℓ=5 **Steps**											
${\#}_{\mathrm{node}}^{\mathrm{obs}}$	$\bar{{\#}_{\mathrm{wh}}\left(0^{+}\right)}$	**Malicious Bots**					**White-Hat Bots**				
		**Step** $10,000^{-}$				**Step** $10,000^{+}$	**Step** $10,000^{-}$				**Step** $10,000^{+}$
		$\bar{{\#}_{\mathrm{mal}}^{\mathrm{obs}}}$	$\bar{{\#}_{\mathrm{mal}}^{\mathrm{in}-\mathrm{obs}}}$	$\bar{{\#}_{\mathrm{mal}}^{\mathrm{unobs}}}$	$\bar{R_{\mathrm{mal}}}$	$\bar{R_{\mathrm{mal}}}$	$\bar{{\#}_{\mathrm{wh}}^{\mathrm{obs}}}$	$\bar{{\#}_{\mathrm{wh}}^{\mathrm{in}-\mathrm{obs}}}$	$\bar{{\#}_{\mathrm{wh}}^{\mathrm{unobs}}}$	$\bar{R_{\mathrm{wh}}}$	$\bar{R_{\mathrm{wh}}}$
10	10	0.00	0.00	0.00	0.00%	0.00%	9.16	81.50	0.17	90.82%	0.17%
20	10	0.00	0.00	0.00	0.00%	0.00%	18.34	72.35	0.13	90.82%	0.13%
	20	0.00	0.00	0.00	0.00%	0.00%	18.15	72.62	0.14	90.91%	0.14%
30	10	0.00	0.00	0.00	0.00%	0.00%	27.47	63.23	0.12	90.82%	0.12%
	20	0.00	0.00	0.00	0.00%	0.00%	27.33	63.46	0.13	90.91%	0.13%
	30	0.00	0.00	0.00	0.00%	0.00%	27.25	63.40	0.10	90.74%	0.10%

**Table 4 sensors-22-09423-t004:** Positioning of this study within the relevant studies.

References	Detection	Mitigation	SpreadPrevention	Disinfection	Observability &Controllability, C&C
Said et al. (2018) [9]		×	×	×	×
Bezerra et al. (2019) [10]		×	×	×	×
Soe et al. (2020) [11]		×	×	×	×
Meidan et al. (2018) [12]		×	×	×	×
Hoang et al. (2018) [13]		×	×	×	×
Mihoub et al. (2022) [14]			×	×	×
Manso et al. (2019) [15]			×	×	×
Ceron et al. (2019) [16]	×		×	×	×
Khattak et al. (2015) [17]	×		×	×	×
US-CERT (2016) [3]	×	×		×	×
Frank et al. (2018) [18]	×	×		×	×
Gopal et al. (2018) [19]	×	×		×	×
Yamaguchi (2020) [23]	×	×	×		×
Pan et al. (2022) [26]	×	×	×		×
This study (2022)	×	×	×

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
