# Peer review of "Botnet Defense System: Observability, Controllability, and Basic Command and Control Strategy"

_sensors, 2022, doi:10.3390/s22239423_

Round 1
Reviewer 1 Report
Introduction section is too long, it should be shorten.
Related work section should present existing approaches, this paper describes BDS in detail, it should be avoided.
Why combining BDS and White-Hat Bot for observability and controllability, what’s the advantage of it?
Author Response
We greatly appreciate your careful review. The following are the answers and corrections to your comments.
Comment (1)
Introduction section is too long, it should be shorten.
Answer and Correction
We reduced the length of the Introduction section to one page.
Comment (2)
Related work section should present existing approaches, this paper describes BDS in detail, it should be avoided.
Answer and Correction
We divided the Related Work section to two sections, Section 2 Related Work and Section 3 BDS, in the new manuscript.
Comment (3)
Why combining BDS and White-Hat Bot for observability and controllability, what’s the advantage of it?
Answer and Correction
We added the advantage of combining BDS with white-hat botnets to the manuscript.
[Revised Manuscript: Page 19, The second paragraph from the end of section 5.5]
IoT systems are open systems. As nodes can be dynamically added or removed, BDS may not be able to observe or control all nodes. This means that observability and controllability are limited and may create cyber security risks in botnet detection and command and control. Combining BDS with white-hat botnets can reduce this limitation. BDS can use self-propagating botnets to indirectly observe and control nodes that BDS cannot directly observe or control through the message link over the botnet. As a result, BDS can detect
and remove botnets even if the network configuration of an IoT system changes.
Reviewer 2 Report
+ In this work, the author proposes an extension of his previous solution called "Botnet Defense System" (BDS) which takes into account Observability and Controllability aspects.
+The paper is well written and well structured. It contains simple examples and good illustrations which make the proposed approach easier to understand.
+ The author shall pay attention to typos and English mistakes (e.g., Line: "Table 2 show the result")
+ The author may include a table which summarizes the works presented in Section 2.
+ In the Related work section, the author needs to talk about works based on Artificial Intelligence for detecting security attacks. They may consider the following reference: https://www.sciencedirect.com/science/article/abs/pii/S0045790622000337
https://www.mdpi.com/1999-5903/10/5/43
https://www.mdpi.com/2073-8994/13/5/866
+ The author validated his approach by means of simulation. It is necessary to explain how this approach will be applied in real-situations.
+ The author also needs to give an estimation of the cost of the adoption of the proposed approach in terms of resource consumption.
+ It is preferable to move proofs (and related figures) into the appendix to make the article to read.
+ The author needs to reduce the number of self-citations.
+ The future directions presented in the conclusion need to be presented in more detail since they are not clear enough.
+ Does the proposed approach cover the situation where the structure of the considered IoT is dynamic (i.e., it evolves over time).
Author Response
We greatly appreciate your careful reviewing. The following are the answers and corrections to your comments.
Reviewer 2’s General Comment
+The paper is well written and well structured. It contains simple examples and good illustrations which make the proposed approach easier to understand.
Reviewer 2’s Comment (1)
+ The author shall pay attention to typos and English mistakes (e.g., Line: "Table 2 show the result")
Answer and Correction
We corrected typo errors, including those you have pointed out.
Reviewer 2’s Comment (2)
+ The author may include a table which summarizes the works presented in Section 2.
Answer and Correction
We summarized the related works in Table 1.
Reviewer 2’s Comment (3)
+ In the Related work section, the author needs to talk about works based on Artificial
Intelligence for detecting security attacks. They may consider the following reference:
https://www.sciencedirect.com/science/article/abs/pii/S0045790622000337
https://www.mdpi.com/1999-5903/10/5/43
https://www.mdpi.com/2073-8994/13/5/866
Answer and Correction
We thank you for introducing us to the excellent paper. We cited all of them in the Related Work section.
[Revised Manuscript: Page 3, The end of the second paragraph of Section 2]
Hoang et al. [15] have proposed a botnet detection model based on machine learning using Domain Name Service query data and confirmed that machine learning algorithms can be used effectively in botnet detection. Mihoub
et al. [16] have proposed an architecture composed of two components for detecting and mitigating DoS/DDoS attacks. The detection component is a multi-class classifier that adopts the Looking-Back concept and identifies
the type of attack and the packet type used in the attack. Based on the result, the mitigation component applies the corresponding countermeasure. Shinan et al. [17] have given a survey on various techniques for detecting botnets, including botnet attacks, machine learning, and Software Defined Networking (SDN). They described graph-based features based on machine learning for bot detection in SDN is the most promising avenue.
Reviewer 2’s Comment (4)
+ The author validated his approach by means of simulation. It is necessary to explain how this approach will be applied in real-situations.
Answer and Correction
We added how to apply our approach in real situations to Section 5.
[Revised Manuscript: Page 10, The first paragraph of Section 5]
Molesky states in Reference [25] that the use of white-hat worm technology can be legally enacted by including explicit terms within the Terms and Conditions agreement at the time of purchase. This suggests that white
hat worm technology is applicable in real situations. We have also implemented a prototype of BDS that defends 24 Raspberry Pi Zero units and have confirmed that our approach works in a real-world environment in this paper.
Reviewer 2’s Comment (5)
+ The author also needs to give an estimation of the cost of the adoption of the proposed approach in terms of resource consumption.
Answer and Correction
We added the influence and cost of introducing and operating the BDS.
[Revised Manuscript: Page 19, The last paragraph of Section 5.5]
The BDS intentionally infects IoT systems with white-hat botnets. This is like a vaccine. The white-hat botnets have the side effect that they can also infect normal devices and consume their resources. However, due to the white-hat worms’ lifespan and the withdrawal strategy , the infection is temporary and has limited impact. The BDS itself requires little cost to introduce and operate. On the other hand, research and development of excellent white-hat botnets, building strategies, and C&C strategies may be costly. By opening up research and development, we expect not only to reduce the cost needed but also to produce innovative results.
Reviewer 2’s Comment (6)
+ It is preferable to move proofs (and related figures) into the appendix to make the article to read.
Answer and Correction
Following your suggestion, we moved all proofs to the appendix.
Reviewer 2’s Comment (7)
+ The author needs to reduce the number of self-citations.
Answer and Correction
We carefully chosen only our important publications and reduced the number of self-citations to less than 15
Reviewer 2’s Comment (8)
+ The future directions presented in the conclusion need to be presented in more detail since they are not clear enough.
Answer and Correction
We gave more concrete directions for the future.
[Revised Manuscript: Page 20, The second paragraph of Section 6]
(i) Resident strategy of white-hat botnets: Malicious botnets will invade IoT systems not just once, but many times. This strategy aims to make a small number of the white-hat bots stay permanently so as to promptly respond to invading botnets at an earlier stage.
(ii) Heterogeneous strategy of white-hat botnets: Like Mirai, white hat botnets have variants with different capabilities. This strategy aims to synergically enhance defense by operating multiple variants in
combination.
(iii) Reinforcement learning-based planning of building and C&C strategies of white-hat botnets: Supervised
learning-based planning of strategies has been proposed in Reference [27] but requires data collection and
labeling before learning. Reinforcement learning works well with multi-agent systems and is expected to
dynamically optimize strategies without human intervention.
(iv) Not only technical but also ethical, legal and social issues (ELSI), empirical testing, and system implementation.
Reviewer 2’s Comment (9)
+ Does the proposed approach cover the situation where the structure of the considered IoT is dynamic (i.e., it evolves over time).
Answer and Correction
Yes. Your comment helps the proposed approach to clarify the benefits of the proposed approach. We added the following as the last paragraph of Section 5.5.
[Revised Manuscript: Page 19, The second paragraph from the end of section 5.5]
IoT systems are open systems. As nodes can be dynamically added or removed, BDS may not be able to observe or control all nodes. This means that observability and controllability are limited and may create cyber security risks in botnet detection and command and control. Combining BDS with white-hat botnets can reduce this limitation. BDS can use self-propagating botnets to indirectly observe and control nodes that BDS cannot directly observe or control through the message link over the botnet. As a result, BDS can detect
and remove botnets even if the network configuration of an IoT system changes.
Round 2
Reviewer 2 Report
The author has addressed all my comments and suggestions. I think the article may be accepted at this stage. Good luck!